# Using a Hand-Held Icterometer to Screen for Neonatal Jaundice: Validation, Feasibility, and Acceptability of the Bili-Ruler^TM^ in Kumasi, Ghana

**DOI:** 10.3390/ijerph22010096

**Published:** 2025-01-12

**Authors:** Ashura Bakari, Ann V. Wolski, Benjamin Otoo, Rexford Amoah, Emmanuel K. Nakua, Jacob Jacovetty, Elizabeth Kaselitz, Sarah D. Compton, Cheryl A. Moyer

**Affiliations:** 1Suntreso Government Hospital, Ghana Health Service, Kumasi AK-039, Ghana; abakari@yahoo.com (A.B.);; 2Department of Emergency Medicine, College of Medicine, University of Cincinnati, Cincinnati, OH 45221, USA; 3Department of Epidemiology and Biostatistics, School of Public Health, Kwame Nkrumah University of Science and Technology, Kumasi AK-039, Ghana; 4Department of Epidemiology, School of Public Health, University of Michigan, Ann Arbor, MI 48109, USA; 5Department of Learning Health Sciences, University of Michigan Medical School, Ann Arbor, MI 48109, USA; 6Department of Obstetrics & Gynecology, University of Michigan Medical School, Ann Arbor, MI 48109, USA; sarahrom@umich.edu

**Keywords:** Bili-Ruler, Ghana, Sub-Saharan Africa, hyperbilirubinemia

## Abstract

Background: Neonatal jaundice (NNJ) remains a leading cause of newborn mortality in much of sub-Saharan Africa. We sought to examine the validity of using a hand-held icterometer as a screening tool to determine which newborns need further assessment. Additionally, we sought to assess the feasibility of its use among mothers. Methods: We recruited and trained healthcare workers at one large district hospital in Ghana to use a hand-held icterometer known as the Bili-Ruler^TM^. We recruited mothers of 341 newborns aged 0 to 2 weeks at the same hospital. Mothers watched a standardized training video, after which they blanched the skin of the newborn’s nose and compared it with the yellow shades numbered one to six on the icterometer. Each newborn was also assessed with a transcutaneous bilirubin meter (TCB). Research assistants and health care workers screened the same newborns, recorded their scores separately, and were blinded to each other’s readings. In the second phase of this study, we recruited 100 new mothers to take the Bili-Ruler home with them, instructing them to check their newborns twice daily. We interviewed them 1–2 weeks later to determine the acceptability and feasibility of its use. Results: Out of 341 newborns screened, 20 had elevated TCB indicative of hyperbilirubinemia. Healthcare workers’ Bili-Ruler ratings had a strong and significant correlation with TCB scores, as did the ratings of researchers and mothers. When comparing Bili-Ruler scores against TCB, sensitivity across all three raters was 80% (95% CI 75.6–84.3), specificity ranged from 61.1% (healthcare providers) to 66.7% (researchers), positive predictive value ranged from 11.4% (healthcare providers) to 13.0% (researchers), and negative predictive value was 98.0% or higher across all raters. Area under the ROC curve ranged from 0.71 for healthcare providers to 0.73 for researchers. Mothers AUC was 0.72. In terms of acceptability and feasibility, the Bili-Ruler was widely accepted by the mothers and family. In total, 98% of mothers reported using it, and 90.8% used it 3 or more days in the first week after birth. Moreover, 89.8% used it more than twice per day. Conclusions: A hand-held, low-tech icterometer is an important potential mechanism for improving early jaundice identification in low-resource settings. Further studies using larger sample sizes with a higher prevalence of hyperbilirubinemia are warranted.

## 1. Introduction

Neonatal hyperbilirubinemia, also known as neonatal jaundice, is a leading cause of death and disability for newborns in low-resource settings, contributing to more than 114,000 deaths each year and leading to long-term disability among another 63,000 newborns [1,2]. Globally, neonatal jaundice has been estimated to contribute to the loss of 4.3 million disability-adjusted life years annually [3].

Neonatal jaundice is a common condition among newborns in the first week after birth. Although jaundice is sometimes associated with prematurity and other health conditions, it often occurs in healthy, full-term newborns [1]. As newborns’ red blood cells break down—which is a normal part of cellular lifecycles—a yellowish waste product called bilirubin is produced. In the early days after birth, many newborns’ livers are not mature enough to keep up with the volume of bilirubin being produced. This can result in elevated levels of bilirubin in the blood, or hyperbilirubinemia, which can be seen through a yellowish tint to the skin and the whites of the eyes, or sclera. Hyperbilirubinemia can lead to bilirubin-related toxicity, which can cause acute, multiorgan-system manifestations and long-term impairments, including irreversible athetoid cerebral palsy (CP) and speech, auditory, and other sensory processing disabilities, as well as death [1,4,5].

Sub-Saharan Africa has the highest incidence of bilirubin-related toxicity in the world, with a rate of 667.8 per 10,000 live births (compared to 4.4 per 10,000 live births in the Americas) [6]. Yet most causes of neonatal jaundice are easily treatable if identified early. Jaundice typically develops in the first few days after birth, after newborns have been discharged home from an institutional birth or in the days following the first post-natal check-up for babies born at home.

Currently, hyperbilirubinemia is assessed in clinical settings using either serum bilirubin (TSB, which requires a blood draw and laboratory testing) or transcutaneous bilirubin measurement (TCB, which requires a light-refracting device that is not readily available in many low-resource settings). TCB has been shown to be a valid predictor of TSB in newborns [7], suggesting that proxies for invasive blood tests are indeed possible.

Researchers from Harvard University developed a low-tech, hand-held icterometer called the Bili-Ruler^TM^ primarily for use in lower-level health facilities without access to TCB or in cases where TSB assessment is not feasible [8] (See Figure 1). The Bili-Ruler has 6 color strips of increasingly yellow hue, digitally standardized and calibrated, to reflect the color of blanched newborn skin of increasing levels of serum bilirubin. Generally a 3 or higher on a scale of 1–6 is indicative of the need for further assessment and potential treatment, whereas a 1 or 2 is indicative of normal levels of bilirubin. The Bili-Ruler—which was validated by researchers against TCB and TSB—has demonstrated high sensitivity and specificity in published trials in the United States and in Bangladesh, yet validation studies to date have only included 63 newborns of African descent [8].

We know that visually judging the changing pigmentation of a dark-skinned infant—especially without a frame of reference—is not straightforward [9]. We also know that neonatal jaundice is most likely to become evident at 3–5 days post birth, when mothers or other caregivers, as opposed to healthcare providers in a clinical setting, are in the best position to assess changes.

In this study, we sought to determine how well visual assessment via Bili-Ruler aligned with TCB assessments among a cohort of newborns in Ghana, as well as whether mothers with newborns at home found the use of the Bili-Ruler to be feasible and acceptable.

Specifically, we aimed to determine: (1) How well healthcare providers’ Bili-Ruler readings compared to TCB assessment, (2) How well different raters (mothers and researchers) agreed with healthcare providers in their use of the Bili-Ruler, as well as how well they agreed with the TCB levels, and (3) How feasible and acceptable did new mothers find the Bili-Ruler when asked to assess their newborns at home for changes in skin pigmentation. The ultimate goal of this research is to determine whether the Bili-Ruler^TM^ could be used by mothers, researchers, or providers as a screening tool to identify neonatal jaundice in low-resource settings, such as Ghana, among dark-skinned infants.

## 2. Materials and Methods

This research consisted of two phases. The first was an observational study comparing a hand-held icterometer to transcutaneous bilirubin assessment, conducted between March and June 2022 at a large district hospital in Kumasi, Ghana, with 341 newborns. The Standards for Reporting Diagnostic Accuracy Studies (STARD) checklist [10] was used to guide methodology and reporting. The second phase (September–November 2022) was an assessment of the feasibility and acceptability of Bili-Ruler use among 100 mothers of newborns who were given a Bili-Ruler to take home for use during the first week after birth.
*Setting*

This research was conducted at Suntreso Government Hospital (SGH), which is situated at the center of Kumasi, the second-largest city in Ghana. Suntreso records an average of 300 monthly births and often receives referrals for both maternal and newborn cases from public and private healthcare facilities in the Ashanti region and beyond. The hospital has a Level II newborn unit with a 34-bed capacity, including the capacity for single and double phototherapy and blood exchange therapy, which are important treatment modalities for neonatal jaundice.

### 2.1. Phase 1: Validation Study

#### 2.1.1. Study Participants

The research included healthcare providers practicing in the mother and baby unit at Suntreso Government Hospital, as well as mothers and their newborns aged 0 to 2 weeks who were randomly recruited based on pre-selected beds. The participating mothers were recruited from three different areas within SGH, including the mother and baby unit, the postnatal outpatient clinic, and post-delivery wards. Mothers with infants newly admitted to the mother and baby unit who were not facing an imminent health crisis were recruited based on randomly selected beds. Mothers with newborns <7 days old presenting at the postnatal outpatient clinic were asked to participate throughout this study period. We also recruited mothers and their newborns from post-delivery wards (based on randomly selected beds) before they were discharged home. Neonates who were already undergoing phototherapy, had exchange blood transfusion performed or were critically ill were excluded from this study. This research also included practicing healthcare providers (doctors, nurses, and physician assistants) who were providing care to the newborns being assessed for jaundice.

#### 2.1.2. Recruitment and Data Collection

Ghanaian research assistants developed a standardized training video for use with both providers and mothers to teach them how to use the Bili-Ruler. All providers in the postnatal outpatient clinics, post-delivery wards, and mother and baby unit were trained in the use of the Bili-Ruler.

Ghanaian research assistants approached the mothers of 341 newborns aged 0 to 2 weeks at Suntreso Government Hospital in Kumasi, Ghana, between March and June 2022. Mothers were taken through an informed consent process and if they agreed to participate, researchers recorded demographic information using Google Forms and a hand-held tablet. Demographic information included maternal age, parity, gestational age, and date and time of delivery. No other identifying information was recorded. Mothers were then shown a Bili-Ruler and the standardized training video. The video demonstrated how to blanch the infant’s nose with the Bili-Ruler and compare the infant’s skin with different yellow shades on the ruler from 1 to 6 (see Figure 1).

Mothers practiced using the Bili-Ruler and were asked to use it on their babies three times in a row, each time reporting the number that most closely corresponded to the color of the blanched skin on the infant’s nose. Assessments were repeated until at least two out of the three measurements yielded the same number, which was taken as the mother’s final score. The trained research assistants repeated the process, recording their own assessments.

A healthcare provider (doctor or nurse) trained in the use of the Bili-Ruler and unaware of the mother’s or research assistant’s score was asked to assess each enrolled newborn using the Bili-Ruler as well. Then a Drager JM-103 transcutaneous bilirubin meter was pressed against the newborn’s forehead, and their transcutaneous bilirubin (TCB) levels were assessed and recorded within one hour of the Bili-Ruler assessment. Researchers recorded the icterometer scores separately for the research assistant, mother, and healthcare workers. Transcutaneous bilirubin (TCB) values were also documented within the Google Form.

#### 2.1.3. Key Variables

Raw Bili-Ruler scores ranged from 1 to 6 for each of the three groups of raters (mothers, researchers, and clinicians). Given that a score of 3 or higher on the Bili-Ruler has been shown by the inventors to be indicative of jaundice warranting further follow-up [8], raw scores were also dichotomized into two groups: normal (Bili-Ruler score of 1 or 2) and elevated (Bili-Ruler score of 3 or higher).

Acceptable bilirubin levels in newborns vary based on gestational age and other factors, thus we used the ‘average risk’ threshold for term babies to calculate estimated TCB threshold for each enrolled newborn, given their age [11]. The average risk threshold was used to create a dichotomous variable indicative of normal versus elevated TCB for each newborn.

#### 2.1.4. Data Analysis: Validation

For our primary outcome, understanding the relationship between healthcare providers’ Bili-Ruler scores and TCB measures, we focused on comparing healthcare providers’ assessments on the Bili-Ruler to the TCB values based on the assumption that healthcare providers would be most familiar with visual assessments for jaundice and thus the closest to a ‘best case scenario’ for accuracy of assessment using the hand-held icterometer. TCB scores were considered the gold standard, against which Bili-Ruler ratings could be compared.

We calculated a Pearson’s R correlation coefficient between the providers’ raw Bili-Ruler scores (rated 1–6) and the original, continuous TCB scores. Pearson’s R is a test to quantify the strength and direction of a relationship between two continuous variables, and it can range from −1 (perfectly negative relationship) to +1 (perfectly positive relationship). A zero would indicate no relationship. Generally, coefficients at 0.7 or higher are considered moderately high correlations.

We then used McNemar’s test to compare providers’ dichotomized Bili-Ruler score (normal vs. elevated) with the dichotomized TCB variable (normal vs. elevated). McNemar’s test is a statistical test that is used for paired categorical data.

For the diagnostic measures of Bili-Ruler as a screening tool for neonatal jaundice, sensitivity and specificity were calculated. Positive and negative predictive values were determined to correctly identify newborns with or without jaundice. The area under the receiver operating characteristic (ROC) curve (AUC) was also calculated to evaluate the overall discriminatory power of the Bili-Ruler as a screening tool.

For our secondary outcome, how well different raters (mothers and researchers) agreed with healthcare providers in their use of the Bili-Ruler, as well as how well they agreed with the TCB levels, we first calculated a Pearson’s R between the raw Bili-Ruler score (rated 1–6) across the three raters (mothers, researchers, and clinicians). We then calculated a Kappa statistic between the groups of raters. The Kappa statistic is a measure of inter-rater agreement when the data are categorical, accounting for the likelihood of agreement occurring by chance. Following this analysis, we used McNemar’s test to compare the level of agreement between raters both in terms of their dichotomous Bili-Ruler assessment (normal vs. elevated) and against the dichotomous TCB variable (normal vs. elevated).

All analyses were conducted using Stata 16.1, and a *p*-value of <0.01 was taken to be significant, yet reported at <0.01 or <0.001 when appropriate.

### 2.2. Phase 2: Feasibility and Acceptability

The second phase of this study involved sending mothers home with Bili-Rulers to determine the feasibility and acceptability of using the hand-held devices at home in the week after birth.

#### 2.2.1. Study Participants

All mothers who had a spontaneous vaginal delivery of a live newborn at Suntreso Hospital and who were ready for discharge home with a live newborn between September and November of 2022 were eligible for participation. Participation required the ability to speak Twi or English. Mothers who were not medically stable or whose babies were not discharged home with them were not eligible to participate.

#### 2.2.2. Recruitment and Data Collection

Similar to the validity study, trained Ghanaian research assistants randomly selected 3 beds from the lying-in ward at the beginning of each week of recruitment. Each morning, mothers in the randomly selected beds were approached and asked about their interest in participating in this research. Those who agreed were taken through an informed consent process, instructed in how to use the Bili-Ruler, given a Bili-Ruler to take home, and instructed to use it twice each day to check their baby’s skin. Given a supply of 100 Bili-Rulers, recruitment was stopped after 100 mothers had agreed to participate.

Preliminary data were collected upon the enrollment of the mothers into this study. This included basic demographic information such as age, education level, and parity. Unique identifiers were assigned to each participant.

On the 5th day after discharge, mothers were called and reminded to bring their newborns for post-natal care, as well as to bring back the Bili-Rulers they had been given. Upon their return, one of the research assistants administered a brief questionnaire based on Weiner et al.’s Acceptability of Intervention Measure (AIM), the Intervention Appropriateness Measure (IAM), and the Feasibility of Intervention Measure (FIM) [12]. Questionnaire data were captured using a Google form.

A subset of mothers was asked an additional set of open-ended questions to better understand the cultural specifics of using the device among the population. Interviews were recorded and transcribed by the research assistants.

#### 2.2.3. Data Analysis: Feasibility and Acceptability

Quantitative data were exported from Google into Stata 16.1 for descriptive analysis. Audio-recorded interviews were transcribed verbatim into Google Docs, ensuring no identifiers were captured. Transcripts were shared with the research team, and 4 of the researchers (A.B., B.O., J.J., and C.A.M.) read all transcripts before discussing and arriving at a preliminary codebook. Coding was conducted by one of the researchers (J.J.), with secondary review by the senior author (C.A.M.). Qualitative codes were compared against quantitative findings using split frame methodology [13].

## 3. Results

### 3.1. Phase 1: Validation Study Results

Table 1 illustrates the demographics of the mothers and newborns enrolled in the validation phase of this study. Mothers had a mean age of 29.5 years; 70% had completed at least some secondary education, and 63.7% had fewer than 3 children. Enrolled newborns averaged 3 days old (range: <1–12 days), with 64.2% being three days old or younger.

Table 2 shows a positive and significant correlation between healthcare providers’ ratings using the Bili-Ruler (scores 1–6) and TCB (r = 0.71, *p* < 0.01). The correlation between research assistants, mothers, and healthcare providers was also positive and significant when compared to one another as well as when compared to TCB.

While correlations explored the relationship between Bili-Ruler scores across all possible options (1–6), Table 3 illustrates the percent agreement across raters when the Bili-Ruler scores were dichotomized to reflect normal levels of bilirubin (scores 1, 2) or a level suggesting elevated bilirubin and the need for further follow-up (scores 3+). The percent of agreement was the highest between research assistants and mothers (88.3% agreement), next highest between mothers and healthcare workers (87.7% agreement), and then between research assistants and healthcare workers (87.1% agreement).

McNemar’s chi square statistics indicate that healthcare workers had significantly different ratings than the research assistants (McNemar’s Chi square = 7.4, *p* = 0.007), but mothers’ ratings were not significantly different from either researchers or healthcare providers. Kappa statistics were significant at *p* < 0.001 for all raters. Kappas ranged from 0.73 to 0.75, indicative of strong agreement across raters.

Table 4 illustrates the comparison between individual raters’ assessments of elevated Bili-Ruler readings (3 or higher) against TCB values that were deemed to be above the threshold indicating the need for jaundice treatment. There were only 20 newborns that were determined to be above the TCB threshold for jaundice treatment. All raters’ Bili-Ruler scores over-estimated the likelihood of needing jaundice treatment: RAs scored 107 newborns as 3+, mothers scored 115 newborns as 3+, and healthcare workers scored 125 as 3+, when TCB indicated 20 newborns as genuinely having elevated bilirubin scores. Positive concordance was seen in 16 out of 20 of the newborns with TCB levels above the threshold. Four out of 20 were misclassified by all raters as not scoring high enough on the Bili-Ruler to warrant follow-up.

Table 5 illustrates the sensitivity, specificity, positive predictive value, negative predictive value, and area under the curve for each of the raters. Figure 2 also illustrates the AUC. Sensitivity assesses how many of those who test positive using the Bili-Ruler are indeed positive, using the TCB values as the gold standard. We found a sensitivity of 80% across all raters, with a 95% CI ranging from 75.6 to 84.3. Specificity refers to how many of those who test negative for jaundice using the Bili-Ruler actually are negative, according to TCB thresholds. We found the lowest specificity among healthcare workers (61.1%), a specificity of 64.2% among mothers, and a specificity of 66.7% among researchers. The positive predictive value (PPV) is the percentage of persons with positive screening results using the Bili-Ruler who actually have jaundice. PPV ranged from 11.4% to 13.0%. Negative predictive value (NPV) is the probability of jaundice being absent if the Bili-Ruler reading is below the threshold. NPV scores were above 98% for all raters. The area under the curve (or AUC) refers to the discriminatory power of the Bili-Ruler to be used for screening, and an AUC of 0.7 or higher is considered good for an exploratory level study [14]. AUC in this case ranged from 0.71 to 0.73.

### 3.2. Phase 2: Feasibility and Acceptability Results

One hundred mothers were recruited into the feasibility portion of our study (see Table 6). Mothers’ mean age was 30 years, with most mothers having a secondary level of education or higher. Most mothers had 2 or more children. Table 7 illustrates the frequencies with which mothers reported using the Bili-Ruler, with 98% saying they used it during the week after birth, 89.9% saying they used it twice per day, and 78.6% reporting they used it twice per day for at least 3 days during the week after birth. Table 7 also illustrates representative quotes from participating mothers regarding their comfort level with using the Bili-Ruler.

Overall, mothers were positively inclined toward the Bili-Ruler, as reflected by quotes such as the following:

“I was using it every day and this morning too”.

“Prevention is better than cure. Having the ruler with you will let you see the jaundice very early if you use it to check. Knowing it early will help in its treatment rather than seeing it too later which will become and problem.”

Table 8 illustrates how many of the 100 mothers were prompted to seek follow-up based on the use of the Bili-Ruler. Out of 98 mothers who used the Bili-Ruler, 26 were prompted to seek care. Of those 26, outcome data were available on 22, where 6 were not jaundiced and sent home, 12 were kept for observation, and 4 were determined to be jaundiced and in need of treatment.

## 4. Discussion

This study found strong inter-rater agreement when healthcare workers, mothers, and researchers were asked to assess newborns using a handheld icterometer, and mothers reported using the Bili-Ruler at home as instructed in the week following their baby’s birth. Mothers also reported appreciating having a tool to help them check their babies for jaundice. Nonetheless, specificity and positive predictive value were not as high as would be expected for a screening tool, although this may be due in part to the low number of true positives in our sample of newborns.

Our findings indicated that, similar to results from Lee et al.’s [8] foundational study of the Bili-Ruler’s measurement properties, Bili-Ruler scores correlated reasonably well with TCB assessments. Lee et al. found a correlation of 0.76, whereas we found a correlation that ranged from 0.67 to 0.77 (mothers had the lowest correlation, healthcare workers were in the middle, and research assistants had the highest correlation, with no significant difference between raters.). Using a cut point of 3 or higher on the Bili-Ruler, Lee et al. reported 91.3% sensitivity and 74.5% specificity for the Bili-Ruler compared to TCB [8]—levels that were higher than what we found in our sample (sensitivity of 80.0% and specificity of 61.1–66.7%). While our sample yielded a higher negative predictive value than Lee et al. (98% vs. 96.1%), positive predictive value was much lower in our sample (11.4–13.0%) than in Lee et al.’s (55.3%). Their sample included a total of 790 newborns, including 390 from Boston (63 of which identified as African American) and 400 from Bangladesh. Thus one potential difference between their findings and ours could be attributable to the darker skinned infants in Ghana that comprised our sample of 341 newborns. In theory, when the skin is blanched, the level of visible yellowing ought to be fairly consistent, regardless of skin tone. However, future research is needed that accounts for differences in skin pigmentation to determine if the Bili-Ruler is equally effective across a variety of skin tones. Another potential factor might be the higher prevalence of elevated bilirubin levels seen in their sample—76 of newborns in Lee’s study (9.6%) had elevated bilirubin levels. This compares to 20 newborns (5.8%) in our sample. It is known that sensitivity can be disproportionately affected in samples with small numbers of true positives, which may explain our finding of 80% sensitivity. We also know that specificity is a function of the number of false positives—and our raters had a tendency to over-estimate the presence of neonatal jaundice, which may explain our 61.1–66.7% specificity findings. We also know that sensitivity and specificity are not as stable in small sample sizes as in larger sample sizes. Thus, our findings should be viewed carefully, as further research with larger sample sizes is necessary.

As mentioned, every type of rater in our study (researchers, mothers, and healthcare providers) overestimated neonatal jaundice using the Bili-Ruler. This needs further exploration to determine if this was a function of observer bias or being primed to look for jaundice and whether more training can correct for this effect or whether this is a problem that may lead to unnecessary return trips to the hospital if mothers were to monitor their newborns from home using a Bili-Ruler. At the same time, we found very few false negatives, which is a desirable trait for a screening test. In this study, it appeared that the Bili-Ruler may over-estimate rather than underestimate the true incidence of jaundice.

This study has several notable strengths. First, researchers used a standardized training video (available in both English and Twi) for the duration of the validation study. This meant all mothers and healthcare providers were instructed to use the Bili-Ruler in the same way. This study also captured data from three different types of raters—healthcare workers, mothers, and researchers—which lends additional face validity to the findings. None of the groups was extremely different from the other, with the exception of healthcare providers scoring lower in their alignment with TCB than researchers and mothers. This suggests that healthcare workers may not be in the best position to visually assess newborns for neonatal jaundice during their busy clinical shifts. Another strength of this study is the inclusion of both quantitative and qualitative data to try to understand the feasibility of using the Bili-Ruler. Our qualitative data contextualized the way mothers used the Bili-Ruler and provided a richness to our understanding we might not have had otherwise. The qualitative data also pointed out erroneous understanding among some of the mothers: even though one mother took her baby to the hospital when the baby’s skin matched #4 on the Bili-Ruler, she had initially thought the numbers on the ruler corresponded to days until her postnatal care visit. Such insights suggest additional training will be necessary if we were to take this intervention to scale.

This study has several limitations, the first of which is the small number of true positive neonatal jaundice cases included in our validation sample. A prevalence of 5.8% was far below what was expected and what is needed to have an accurate positive predictive value. Future research with larger sample sizes that include more true positives is warranted. It will also be important to conduct robust power analysis that accounts for the number of cases of neonatal jaundice identified. Another limitation relates to the potential for bias in assessments because raters cannot be effectively blinded to their own previous scores. While we do not think this unduly influenced our findings, it is possible that raters tried to match their first score rather than objectively treating each observation as unique. Future studies using the Bili-Ruler could consider creative ways to minimize the risk of bias, including having providers assess multiple babies at once and then repeat the assessments in a different order.

## 5. Conclusions

These findings have several implications for research and practice. First, it is clear that mothers appreciated the ability to take something home with them to allow them to check their baby’s health. The current care paradigm in Ghana and elsewhere requires mothers to come to a health facility for newborn monitoring, when it is possible that low-tech devices like the Bili-Ruler could provide opportunities to empower mothers and families for home-based monitoring of their newborn. It is also possible that the Bili-Ruler could be used by community health nurses and community health centers in peripheral areas, as few have the opportunity to screen for jaundice any other way. However, before such devices can be deployed at scale, further validation research and more real-world effectiveness testing is required.

In summary, these findings show promising evidence for the potential utility of a low-cost, hand-held icterometer as one way to improve recognition of neonatal jaundice in low-resource settings. However, further validation research is necessary to ensure that the Bili-Ruler’s measurement properties are consistent, regardless of race and skin pigmentation.

## Figures and Tables

**Figure 1 ijerph-22-00096-f001:**
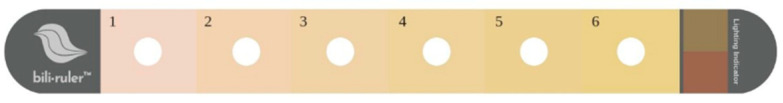
Hand-held Bili-Ruler^TM^ for monitoring jaundice; newborn’s nose is lightly blanched with the Bili-Ruler, and blanched skin is viewed through the hole in the center of the color swatch for color comparison. A 3 or higher is indicative of the need for additional screening.

**Figure 2 ijerph-22-00096-f002:**
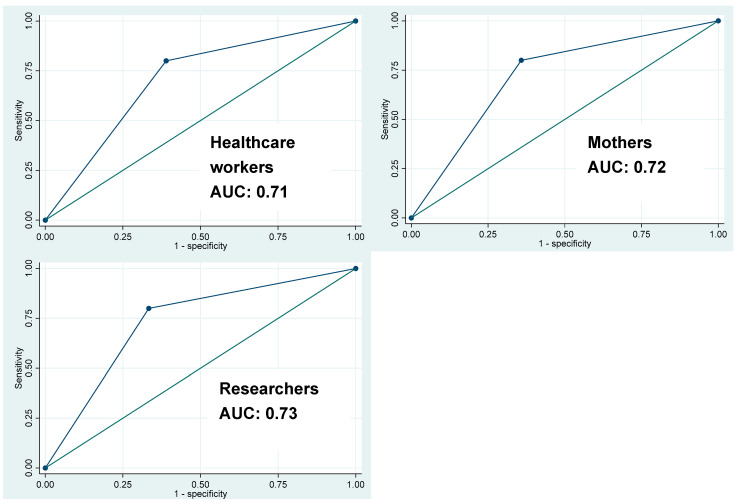
Area under the ROC curves for each type of rater’s Bili-Ruler score when compared to TCB score.

**Table 1 ijerph-22-00096-t001:** Participant demographics (N = 341).

Variable	Mean (95% CI)
Mean maternal age (years)	29.5 (28.9, 30.1)
Parity	2.4 (2.2, 2.6)
Baby’s age (days)	3.0 (2.7, 3.3)
	**Frequency (%)**
Maternal age in categories (years)	
15–25	74 (21.7)
26–35	209 (61.3)
36+	58 (17.0)
Maternal education	
No education	18 (5.3)
Primary	20 (5.9)
Secondary	238 (70.0)
Tertiary	64 (18.8)
Parity	
1	124 (36.4)
2	93 (27.3)
3	57 (16.7)
4	28 (8.2)
5+	39 (11.4)
Baby’s age (days)	
<1	72 (21.1)
1–3	147(43.1)
4–6	62 (22.9)
7–12	44 (12.9)

**Table 2 ijerph-22-00096-t002:** Correlation matrix across raters and TCB scores.

	Research Assistant	Mother	Healthcare Worker	TCB Value
Research assistant	1.0			
Mother	0.76 *	1.0		
Healthcare worker	0.82 *	0.78 *	1.0	
TCB value	0.77 *	0.67 *	0.71 *	1.0

* *p* < 0.01.

**Table 3 ijerph-22-00096-t003:** Percent agreement and interrater reliability for an assessment of 3 or higher on the Bili-Ruler^TM^.

	Researcher	Mother	Healthcare Worker
Researcher	-		
Mother	% agreement: 88.3 McNemar’s Chi^2^: 1.6 *p*-value: 0.21 Kappa: 0.75 *p*-value: <0.001	-	
Healthcare worker	% agreement: 87.1 McNemar’s Chi^2^: 7.4 *p*-value: 0.007 Kappa: 0.73 *p*-value: <0.001	% agreement: 87.7 McNemar’s Chi^2^: 2.4 *p*-value: 0.12 Kappa: 0.74 *p*-value: <0.001	-

**Table 4 ijerph-22-00096-t004:** Comparison of rater’s assessment of elevated Bili-Ruler readings against TCB threshold.

**Research Assistant**	**Elevated TCB**		**Total**
	No	Yes	
Bili-Ruler normal (1, 2)	214 (66.7%)	4 (20.0%)	218 (63.9%)
Bili-Ruler elevated (3+)	107 (33.3%)	16 (80.0%)	123 (36.1%)
	321	20	341
		Chi^2^ = 17.8, *p* < 0.001	
**Mother**	**Elevated TCB**		**Total**
	No	Yes	
Bili-Ruler normal (1, 2)	206 (64.2%)	4 (20.0%)	210 (61.6%)
Bili-Ruler elevated (3+)	115 (35.8%)	16 (80.0%)	131 (38.4%)
	321	20	341
		Chi^2^ = 15.5, *p* < 0.001	
**Healthcare Worker**	**Elevated TCB**		**Total**
	No	Yes	
Bili-Ruler normal (1, 2)	196 (61.1%)	4 (20.0%)	200 (58.7%)
Bili-Ruler elevated (3+)	125 (38.9%)	16 (80.0%)	141 (41.4%)
	321	20	341
		Chi^2^ = 13.1, *p* < 0.001	

**Table 5 ijerph-22-00096-t005:** Sensitivity, specificity, positive predictive value, negative predictive value, and area under the curve for the Bili-Ruler vs. TCB by rater.

	Healthcare Providers % (95% CI)	Mothers % (95% CI)	Researchers % (95% CI)
Sensitivity	80.0% (75.6–84.3)	80.0% (75.7–84.3)	80.0% (75.8–84.3)
Specificity	61.1% (55.9–66.2)	64.2% (59.1–69.3)	66.7% (61.7–71.7)
Positive predictive value	11.4% (8.0–14.7)	12.2% (8.7–15.7)	13.0% (9.4–16.6)
Negative predictive value	98.0% (96.5–99.5)	98.1% (96.6–99.6)	98.2 (96.7–99.6)
AUC	0.71 (0.61–0.80)	0.72 (0.63–0.81)	0.73 (0.64–0.82)

There were no statistically significant differences between raters.

**Table 6 ijerph-22-00096-t006:** Participant demographics of feasibility study (N = 100).

	Mean, (SD)
Newborn’s age at birth (weeks)	39.4, (0.3)
Mother’s age (years)	30.0, (0.6)
Mother’s education level	n, (%)
None	2, (2.0)
Primary	31, (31.3)
Secondary	49, (49.4)
Tertiary	17, (17.2)
Parity	
1	28, (28.0)
2–3	51, (51.0)
4+	21, (21.0)

**Table 7 ijerph-22-00096-t007:** Use and attitudes towards the Bili-Ruler^TM^.

Use of Bili-Ruler		n (%)/Representative Quotes	
Number who used Bili-Ruler		98 (98.0)	
Used Bili-Ruler twice per day or more		88 (89.8)	
Number of days used during week after birth (days)		1–2	9 (9.3)
		3–5	50 (51.6)
		6+	38 (39.2)
Number of days used twice during week after birth		0	11 (11.2)
		1–2	10 (10.2)
		3–5	68 (69.4)
		6+	9 (9.2)
Was the Bili-Ruler™ easy to use?		“I was having some difficulties using it on the first day, I wasn’t all that sure about the color because my baby was resisting when I put on her nose but from day 2 going, I became comfortable and I used it well to know the color which matches the skin. Yes, I used it and it’s very helpful” “I think it is possible for every woman to use because it’s not complex”	
No	10, (10.2)		
Yes	88, (89.8)		
Was it difficult to determine the difference between the colors on the Bili-Ruler™?		“Sometimes we find it difficult in detecting the yellowish color on our babies, so with this (bili) ruler we can easily see the type of yellow which means danger on our baby’s skin”	
No	80, (81.6)		
Yes	18, (18.4)		
Confidence in matching the color on the Bili-Ruler to the baby’s skin		“Haha…to be pretty sure of the color which matches the nose, I had to check it about 3 times to see if it’s really matches. I open my curtains to get the outside light and check it well. I was comfortable with the color I was seeing” “I will give it 80% sure of the color. Sometimes I had to do it twice to really see which matches”	
Did not know	1, (1.0)		
Pretty sure but not positive	29, (29.6)		
Very confident	68, (69.4)		
Reason for not using the Bili-Ruler		“It will be difficult to tell but I think when we are overwhelmed with other house duties, we turn to forget some petty things, hence we might not use. Aside this I don’t think there will be any issues unless sometime personal…” “Well some people turn to rely their beliefs. Someone might think her baby has been seen by someone with bad eye that’s why their baby’s skin color or eye being yellow. They will then try and use some herbs and be putting the baby under the sun for some period and when it’s not resolving that’s when they will come to the hospital but by the, it might be late for her. People have different ways of understanding things, so when you give the ruler to such person, she will not use it”	
Forgot to use it	28, (28.0)		
Lost or left it somewhere	2, (2.0)		
Too busy to use it	33, (33.0)		
Too hard to use	6, (6.0)		
Did not want to use it	1, (1.0)		

**Table 8 ijerph-22-00096-t008:** Outcome of Bili-Ruler use.

Variable	N (%)	Illustrative Quote
Prompted care-seeking		“When I got home, I started using it the next 2 days, so in the morning, I took it to check my baby. Upon the video I watched and how you told me to use it, I forgot, so I thought each day was assigned to each number, so say day 1 was for color 1 and day 2 was for color 2 and so on. Fortunately for me on day 5, I realized the eye is becoming yellow like color 4 on the ruler, so I decided to come to the hospital”
Yes	26, (27.0)	
No	71, (73.2)	
Result of care-seeking		“The ruler has different shades of yellow colors which will help us identify the bad ones. See we mostly want to see the baby’s body color has changed to a deep yellowish before we know it’s jaundice, but through this education and the ruler we can see it early and seek for treatment at the right time. This happened to me with my first born, I didn’t know which yellow color is bad so I realized the baby’s eye was very yellowish on the 5th day after birth, that was when I brought him to the hospital. So since then, I have learnt a lot about this jaundice and you adding the ruler will really help we the mothers”
Not jaundiced, sent home	6, (27.3)	
Kept for observation	12, (54.5)	
Jaundiced	4, (18.2)	

## Data Availability

Data are available upon reasonable request from the corresponding author. Data have not been deposited into a public repository due to privacy concerns.

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
