# Peer review of "Using a Hand-Held Icterometer to Screen for Neonatal Jaundice: Validation, Feasibility, and Acceptability of the Bili-Ruler^TM^ in Kumasi, Ghana"

_ijerph, 2025, doi:10.3390/ijerph22010096_

Round 1
Reviewer 1 Report
Comments and Suggestions for Authors
Although small in scale, this prospective cohort study examined the usefulness of a portable icterometer in screening for pathological neonatal jaundice. The reliability of the principle was verified in phase 1, and the high correlation with TCB and the agreement rate between the mother, researcher, and medical professional proved that it was an objective screening method. What is excellent about this paper is that it examines whether it is possible to use it in clinical settings in phase 2. Although it is not statistical proof, Tables 7 and 8 are considered sufficient proof for this paper's scale. The paper is designed in such a way that it is very informative, with actual opinions also included in Tables 7 and 8. The discussion of the results is detailed, with a discussion of the low positive predictive value in L336-340 and the high negative predictive value in L348-349. Please submit an early corrected manuscript after making the following corrections.
1) For the majority of readers who have never actually seen the Bili-Ruler, this is a research method that is difficult to imagine. It would be easier to understand if you included a photo or illustration of the Bili-Ruler and clearly explained how to use it.
2) The power analysis for the number of cases has yet to be considered. Adding this to the limitations section of the discussion would further improve the quality of the paper.
Author Response
Comment 1) For the majority of readers who have never actually seen the Bili-Ruler, this is a research method that is difficult to imagine. It would be easier to understand if you included a photo or illustration of the Bili-Ruler and clearly explained how to use it.
Response 1) Thank you for this suggestion. Figure 1 was added on Page 4 to address this concern.
Comment 2) The power analysis for the number of cases has yet to be considered. Adding this to the limitations section of the discussion would further improve the quality of the paper.
Response 2) Thank you for this suggestion. This was added to the discussion section.

Reviewer 2 Report
Comments and Suggestions for Authors
Thank you for the opportunity to review such an interesting and well-organized work, with regard to a very fragile clinical issue, such as rapid identification of hyperbilirubinemia, which might prove to be really important especially in low-resource healthcare settings. This is a well-written and well-designed study, and should definitely be published.
A few minor remarks for authors:
1) Line 147: Please define what RA score is for less experienced reader on the topic.
2) Could you please provide a rough picture of the Bili-Ruler in the Materials and Methods section? That would give a better overall picture to readers.
3) Table 5: With regard to table 5, you should probably highlight in the Discussion section that differences between groups were not statistically significant, and perhaps indicate it on the table.
4) Lines 337-338: Would you suggest a different Bili-Ruler coloring for darker-skinned infants?
I am looking forward to a future study, addressing the issues of bias leading to the lower positive predictive value of the Bili- Ruler in the present study.
Reviewer 3 Report
Comments and Suggestions for Authors
Dear authors, thank you for the submission of your manuscript.
This work is highly valuable for the screening of hyperbilirubinaemia in areas where infrastructure and medical aid is less advanced, than in other regions of the world.
However, I have a few remarks, that could help to improve the already high quality of the manuscript a bit:
1. For the readers with less medical and biological background I would suggest to describe the etiology of neonatal jaundice. I know that it is merely textbook knowledge, but still I think it will be worthwhile to include it.
2. In the materials and methodes section, a short description of the 3 used methodes (McNemar`s chi square, Kappa statistics, and Pearson`s R correlation) would help the readers, that are not familiar with statistical methods to understand the results.
The authors should also explain why the use different p-values for the definition what is significant (<0.001 and <0.01). But on page 4, lines 190-191 the authors claim that for all analyses a p-value of <0.01 was taken as significant. This needs to be clarified and/or corrected.
3. Table 4: In the field Mother/Resaercher there is a p-value of <0.21 for the McNemar`s Chi2, but this could mean any p-value between 0.20 and < 0.001, so it is not defined whether it is significant or not significant.
4. Just a typo on page 12, line 325 "Lee et al.`s"
Reviewer 4 Report
Comments and Suggestions for Authors
The authors examine a billi-ruler to examine bilirubin levels in newborn babies. The test was performed by health workers, mothers and by nurses. It should be explained why specifity and sensitivity have a value of about 80 %. The description of the performing of the test was good, and the presentation in tables is understandable. As far as I can judge, the English language is good enough. The list of references could be enlarged. This is an important research article.
